# Integrated Control of Fatty Acid Metabolism in Heart Failure

**DOI:** 10.3390/metabo13050615

**Published:** 2023-04-29

**Authors:** Xiaoting Li, Xukun Bi

**Affiliations:** Key Laboratory of Cardiovascular Intervention and Regenerative Medicine of Zhejiang Province, Department of Cardiology, Sir Run Run Shaw Hospital, Zhejiang University School of Medicine, Hangzhou 310016, China; lixiaoting26@zju.edu.cn

**Keywords:** fatty acid homeostasis, heart failure, cardiac lipotoxicity

## Abstract

Disrupted fatty acid metabolism is one of the most important metabolic features in heart failure. The heart obtains energy from fatty acids via oxidation. However, heart failure results in markedly decreased fatty acid oxidation and is accompanied by the accumulation of excess lipid moieties that lead to cardiac lipotoxicity. Herein, we summarized and discussed the current understanding of the integrated regulation of fatty acid metabolism (including fatty acid uptake, lipogenesis, lipolysis, and fatty acid oxidation) in the pathogenesis of heart failure. The functions of many enzymes and regulatory factors in fatty acid homeostasis were characterized. We reviewed their contributions to the development of heart failure and highlighted potential targets that may serve as promising new therapeutic strategies.

## 1. Introduction

Heart failure is a complex clinical syndrome characterized by elevated intracardiac pressure or inadequate cardiac output due to structural or functional cardiac abnormalities [1]. It is a major cause of death in patients with heart disease. Coronary artery disease and hypertension are the predominant factors leading to heart failure [2]. The large improvement in healthcare worldwide has failed to improve the five-year mortality rate of heart failure, which remains at 52.6% [3]. Therefore, there is an urgent need to discover new therapeutic targets for the treatment of heart failure.

A normal heart predominantly relies on fatty acid oxidation (FAO) to produce energy; however, a failing heart is characterized by a decrease in FAO [4,5]. Fatty acids (FAs) are carboxylic acids that contain hydrocarbon chains, ranging from 4–36 carbon atoms. Based on the number of carbon atoms, FAs can be divided into four groups: short-chain FAs (up to 6 carbon atoms), medium-chain FAs (8–12 carbon atoms), long-chain FAs (14–18 carbon atoms), and very long-chain FAs (above 20 carbon atoms). The chains of FAs are either fully saturated (do not contain double bonds) or contain one or more monounsaturated or polyunsaturated double bonds. The most commonly occurring FAs contain an even number of carbon atoms with lengths of 12–24 carbon atoms (Table 1). The heart prefers to use long-chain fatty acids as oxidizing substrates to maintain normal function [6]. The utilization of FAs requires complex metabolic processes. Alterations in the key enzymes involved in this process result in aberrant FA metabolism, which contributes to the severity of heart failure.

This review summarized recent knowledge regarding advances in FA metabolism during heart failure. We discussed the function of each key enzyme involved in the regulation of FA uptake, lipogenesis, lipolysis, and FAO in heart failure. We also examined the transcriptional and phosphorylation regulation of FA homeostasis. Based on these results, this is the first review to explore the potential roles of enzymes and regulatory proteins, which might be considered in the treatment of heart failure.

## 2. Fatty Acid Uptake

Cells obtain FAs via de novo lipogenesis and exogenous uptake (Figure 1). The entry of exogenous FAs from the surrounding microenvironment into cells is facilitated by specific transporters, including fatty acid translocase (FAT/CD36), fatty acid transport protein family (FATPs), and heart-type fatty acid-binding proteins (H-FABP) [6]. 

### 2.1. Fatty Acid Translocase (FAT/CD36)

Fatty acid translocase (FAT) was discovered in rat adipocytes as an 88 kDa integral membrane protein participating in FA transport. The primary protein sequence is 85% homologous to glycoprotein IV (CD36) [7]. FAT/CD36 is abundantly expressed in the heart, intestine, fat, and muscle. Several studies reported a high prevalence of FAT/CD36 deficiency in patients with dilated [8], ischemic [9], or hypertrophic cardiomyopathy with asymmetric septal hypertrophy [10]. Clinical studies in the Japanese population show that point mutations in FAT/CD36 (^478^C>T) may result in deficiency [11,12]. Similarly, patients with heart failure have decreased FAT/CD36 and overall FA content, whereas FAT/CD36 gene expression is elevated after left ventricular assist device implantation, although the FA content remained unchanged [9]. These studies imply that FAT/CD36 deficiency might be an etiology of heart failure.

Animal studies conducted over the last two decades further clarified the function of FAT/CD36 in heart failure. The myocardial FAT/CD36 expression level correlates with the cardiac ejection fraction in the infarcted rat heart, with a 36% reduction in the protein level compared to sham-operated controls [13]. FAT/CD36 decreases in hypertrophic hearts in response to pressure overload. Tamoxifen-inducible cardiomyocyte-specific CD36 knockout mice exhibit rapid progression from compensated hypertrophy to heart failure [14]. These phenotypes may be caused by sufficient adenosine triphosphate (ATP) production and increased glycolytic flux-mediated structural remodeling [15]. However, FAT/CD36 is highly induced in cardiac lipotoxicity models, including age-induced murine cardiomyopathy and diabetic cardiomyopathy. FAT/CD36 deficient mice show significantly enhanced cardiac function, with lower intramyocardial lipid levels and improved ATP production [16]. Similarly, FAT/CD36 deficient mice show reduced cardiomyocyte triacylglycerol (TAG) accumulation and cardiac dysfunction after high-fat diet feeding [17]. These findings strongly support that moderate levels of FAT/CD36 protect against hemodynamic stress, whereas high expression is associated with cardiac lipotoxicity.

### 2.2. Fatty Acid Transport Protein Family (FATPs)

Fatty acid transport proteins facilitate the transmembrane transport of FA [18]. These candidate proteins were firstly identified by cloning an adipocyte cDNA library in COS7 cells and exhibiting an uptake of fluorescent FA analogs. This protein was named FATP1. FATP1 is expressed in most mammalian tissues, including the heart, brain, adipocytes, and kidney. Cardiac-specific overexpression of FATP1 using the α-myosin heavy chain gene promoter increases cardiomyocyte FA accumulation and stimulates FAO. However, the heart predominantly exhibits diastolic dysfunction and a prolonged corrected QT interval (QTc) after three months. These results confirm the role of FATP1 in FA import into the heart [19]. Another predominant cardiac FATP is FATP6, which is over 20 times more abundant than FATP1 in mouse heart lysates. The heart preferentially takes up palmitate compared to oleate, and the FATP6-stable cell line has the same uptake pattern. The authors suggest that a significant portion of FA uptake in the heart is mediated by FATP6 [20]. A clinical study revealed that a variation in the 5′-untranslated region of the FATP6 gene (^7^T>A polymorphism) is associated with features of metabolic syndrome and signs of myocardial alteration or heart failure [21]. The protein levels of FATP6 and FATP1 decrease in infarcted heart failure mice [13]; however, the detailed mechanism of reduced FATPs is less clear in these animal models.

### 2.3. Heart-Type Fatty Acid-Binding Protein (H-FABP)

Fatty acid-binding proteins are low-molecular-weight proteins (approximately 15 kDa) that facilitate FA uptake. Heart-type FABP (H-FABP) is probably the best-known member of the FABP family. It is encoded by FABP3, which is located in the 1p33-p32 region of chromosome 1 [22]. Tissues with a high demand for FAs (including the heart, brain, kidney, and adrenal gland) express H-FABP. H-FABP is more abundantly expressed in the ventricle and atrium than in other organs [23]. It can be rapidly released from myocytes into circulation, owing to its small molecular weight and free cytoplasmic localization; therefore, it is thought to be a biomarker in the pathogenesis of heart failure [24]. H-FABP levels are significantly increased in patients with chronic heart failure [25], which correlates with the New York Heart Association (NYHA) functional class and inversely correlates with ejection fraction. Furthermore, persistently high H-FABP levels are associated with adverse events during the follow-up of patients with heart failure [26]. Mice with disrupted H-FABP show elevated plasma long-chain FA (LCFA) levels, decreased cardiac deposition of an LCFA analog, and increased cardiac deoxyglucose uptake; therefore, there is a requirement for H-FABP in cardiac LCFA utilization [27]. Furthermore, glucose oxidation increases in H-FABP-knockout mice to compensate for cardiac energy production, as the palmitate uptake and oxidation are reduced [28]. 

## 3. Lipogenesis

Lipogenesis is the process of FA synthesis from non-lipid precursors and mainly occurs in the liver and adipose tissues. Fatty acids are synthesized in the heart using several lipogenic enzymes [29]. Lipogenesis encompasses two major processes: the activation of acetyl-coenzyme A (acetyl-CoA) and FA biosynthesis. The main substrate for lipogenesis is acetyl-CoA, which is derived from acetate by acetyl-CoA synthetase (ACS) or citrate by ATP-citrate lyase (ACLY). The carboxylation of acetyl-CoA to malonyl-CoA by acetyl-CoA carboxylase (ACC) is an irreversible rate-limiting step, and the opposite reaction is catalyzed by malonyl-CoA decarboxylase (MCD). Next, seven molecules of malonyl-CoA and one molecule of acetyl-CoA are condensed by fatty acid synthase (FASN), which finally produces saturated palmitate. Palmitate desaturation by stearoyl-CoA desaturase (SCD) generates monounsaturated FA (MUFA). Polyunsaturated FAs (PUFAs) are formed after a series of elongation and desaturation steps (Figure 1) [30]. 

FA membrane transport proteins are supposed to induce endogenous lipogenesis by increasing FA uptake into the heart. However, muscle-specific overexpression of FAT/CD36 failed to show significant changes in cellular lipids, such as TAG and phospholipids [31]. Consistently, overexpression of FATP1 or H-FABP in the heart did not affect the cardiac TAG content [19,28]. These results suggest that the activation of lipogenesis may require integrated control of particular enzymes.

### 3.1. Acetyl-CoA-Producing Enzymes ACS and ACLY

Acetyl-CoA synthetase (ACS) produces acetyl-CoA via the ligation of acetate and CoA, and acetyl-CoA is utilized to synthesize FAs in an anabolic pathway or via mitochondrial oxidation via a catabolic pathway. The series of reactions begins with acetyl-CoA; therefore, the acetyl-CoA level is a key element in FA metabolism. The ACS family comprises a large group of enzymes subdivided into five subfamilies: short-chain ACS (ACSS), medium-chain ACS (ACSM), long-chain ACS (ACSL), long-chain synthetase (ACSVL), and bubblegum ACS (ACSBG) [32]. 

Long-chain ACS is the most intensively studied enzyme in the ACS family, owing to its essential role in cardiac function. It participates in FA thioesterification to produce long-chain fatty acyl-CoA, together with acetyl-CoA production [33]. The five identified ACSL isoforms (ACSL1, ACSL3-6) exhibit specific substrate preferences, and responses to nutritional and hormonal regulation. ACSL1 is the major isoform expressed in highly oxidative tissues, including the heart, skeletal muscle, and brown adipose tissues. Mice lacking cardiac ACSL1 show impaired FAO and cardiac hypertrophy [34]. However, studies using ACSL1 transgenic mice have exhibited opposite results. For example, cardiac overexpression of ACSL1 leads to cardiac lipotoxicity, resulting in mild left ventricular hypertrophy with modest systolic dysfunction, which is associated with increased production of reactive oxygen species (ROS) and FA uptake, followed by mitochondrial remodeling [35]. In contrast, cardiac dysfunction induced by transverse aortic constriction (TAC) is improved in ACSL1 transgenic hearts by reducing cardiac lipotoxicity [36]. 

ATP citrate lyase (ACLY) is a cytosolic enzyme that generates acetyl-CoA for fatty acid biosynthesis. Citrate was used as the substrate for ACLY, unlike ACS. It is derived from pyruvate oxidation in the tricarboxylic acid (TCA) cycle (which is derived from glucose) [37], or through reductive carboxylation (which is derived from glutamine) [38]. Therefore, ACLY represents a combination of three different metabolic pathways. Homozygous ACLY knockout mice died early in development, and heterozygous mice were healthy, fertile, and normolipidemic on both chow- and high-fat diets. Half-normal amounts of ACLY in heterozygous mice did not perturb triglyceride or cholesterol synthesis or ACS expression [39]. Genetic variants in genes encoding ACLY are associated with the risk of cardiovascular events [40]; however, the role of ACLY in heart failure remains to be explored. 

### 3.2. Fatty Acid Biosynthesis Enzymes

Acetyl-CoA carboxylase (ACC) and malonyl-CoA decarboxylase (MCD) are responsible for malonyl-CoA production. The heart predominantly relies on FAO to fuel ATP production, and the FAO flux is regulated at the level of entry of long-chain fatty acyl-CoA into the mitochondria by carnitine acyltransferase I (CPT1). Malonyl-CoA is a CPT1 inhibitor, which is essential for regulating cardiac function. Cardiac contractile stimulation activates MCD, which decreases the malonyl-CoA content and, hence, increases FAO [41]. MCD levels are highly induced in high-fat diet-fed rats and streptozotocin-treated diabetic model. Peroxisome proliferator-activated receptor-α (PPARα) is activated by the increasing concentrations of non-esterified FAs in the plasma, and thereby stimulates the expression of MCD. In contrast, pressure overload-induced cardiac hypertrophy has reduced MCD activity owing to PPARα inhibition. This results in the suppression of FAO [42]. Notably, pharmacological inhibition or genetic deletion of MCD protects the ischemic heart by inhibiting FAO and stimulating glucose oxidation [43,44,45]. 

In contrast, acetyl-CoA activates ACC; this results in increased malonyl-CoA levels that inhibit FAO [46]. The two main ACC isoforms in mammals (ACC1 and ACC2) have distinct tissue distributions and functions: ACC1 is predominantly expressed in the liver and adipocytes, whereas ACC2 is enriched in the heart and skeletal muscle [47]. Furthermore, ACC1 is localized in the cytosol, and ACC2 is associated with the mitochondria [48]. Mice with a null mutation in ACC1 show embryonic lethality, while ACC2-null mice are viable [49]. ACC2 knockout mice have a normal lifespan, a higher rate of FAO, and a lower amount of fat compared to wild-type mice [50]. Moreover, ACC2 mutant hearts display normal functional parameters despite a significant decrease in size, with a marked preference for the oxidation of glucose and Fas [51]. Cardiac-specific deletion of ACC2 leads to a reduction in cardiac malonyl-CoA with an increase in FAO and decreased glucose utilization. An ACC2 knockout in the heart did not affect left ventricular function or oxygen consumption. A genetic deletion of ACC2 attenuates cardiac hypertrophy and prevents metabolic remodeling during pressure overload hypertrophy [52]. Furthermore, an ACC2 deletion with pre-existing cardiac pathology improves cardiac function during the transition from pathological hypertrophy to heart failure, but only in female mice. This different response derives from a sex-dependent regulation of the PPARα signaling pathway in heart failure. ACC2 knockout mice resist obesity and retain insulin sensitivity in a high-fat diet-induced diabetes model [49]. This is thought to protect the heart from diabetic cardiomyopathy.

### 3.3. Fatty Acid Synthase (FASN) and Stearoyl-CoA Desaturase (SCD)

Fatty acid synthase is a multi-enzyme complex that catalyzes the conversion of acetyl-CoA and malonyl-CoA to long-chain saturated FAs. Heterozygous FASN mutants are ostensibly normal; however, the null FASN mutant results in embryonic lethality [53]. FASN also generates an endogenous ligand for PPARα signaling [54,55]. It is highly induced in murine heart failure models, including ACS transgenic mice. Moreover, the mRNA level of FASN increases in cardiac tissue from individuals with sudden cardiac death and in patients with class IV cardiomyopathy requiring left ventricular assist device (LVAD) support. Cardiac specific FASN knockout mice manifested normal resting heart function, cardiac PPARα signaling, and FAO. However, most mice died within one hour of pressure overload induced by transverse aortic constriction, probably due to arrhythmia. Calcium–calmodulin (CaM)-dependent protein kinase II (CaMKII) signaling appears to be pathogenic because inhibition of this signaling rescues mice from early mortality after transverse aortic constriction [56].

Stearoyl-CoA desaturase is the rate-limiting enzyme that catalyzes the synthesis of MUFA, such as palmitoleic acid and oleic acid, which are substrates for the synthesis of phospholipids, cholesterol, and triacylglycerol. Four isoforms (SCD1–4) are identified in mice, and SCD1, SCD2, and SCD4 are expressed in the heart. SCD1 and SCD5 were identified in humans, and each isoform is found in the heart. Compared with wild-type mice, the heart tissues from SCD1 knockout mice have reduced FA transport and oxidation, as well as increased glucose uptake and oxidation. The left ventricular weight is higher in SCD-deficient mice; however, cardiac function is not significantly affected [57]. In addition, there is no significant difference in SCD activity and palmitoleic acid levels in SCD1 knockout mice, as SCD4 compensates for the lack of SCD1 in the heart [58]. Human plasma SCD activity is also associated with the risk of heart failure [59]. SCD1 expression is induced in several heart failure models [60]. The myocardium-specific expression of SCD1 mice triggers cardiac hypertrophy and symptoms of heart failure at an age of eight months with an overload of cardiotoxic saturated lipids [61]. The loss of SCD1 by RNAi still led to cardiac dysfunction and lipotoxicity [62]; however, systemic SCD1 deficiency in ob/ob obese mice improves impaired cardiac function owing to the inhibition of apoptosis [63]. Together, these findings suggest that SCD1 participates in pathological remodeling during heart failure.

### 3.4. Diacylglycerol Acyltransferase (DGAT) 

Diacylglycerol acyltransferase catalyzes the final step in the formation of triglycerides from diacylglycerol (DAG) and fatty acyl-CoA. The mammalian heart has two DGAT isoforms: DGAT1 and DGAT2. Failing human hearts show severely reduced DGAT1 expression with the accumulation of DAG and ceramides [64]. DGAT1-null mice show no cardiac phenotype, with normal TAG and DAG levels in the heart [65]. However, the cardiac-specific deletion of DGAT1 increases DAG levels without altering TAG levels. Half of the cardiac DGAT1 knockout mice died by nine months of age, with an increase in heart failure markers and cardiac dysfunction [66]. In contrast, transgenic overexpression of DGAT1 in the heart results in approximately normal cardiac function at three to four months of age. ACS and DGAT1 double-transgenic mice show improved heart function compared to ACS transgenic mice [64]. However, DGAT1 transgenic mice develop severe cardiomyopathy with moderate systolic dysfunction at 52 weeks of age [67]. DGAT1 knockout mice are resistant to obesity and insulin without changes in TAG levels in a high-fat diet-induced murine model. Co-inhibition of DGAT1/2 in the cardiomyocytes still protects the heart against high-fat diet-induced lipid accumulation and decreases TAG levels [68]. Recent data suggest that DGAT plays an important role in the pathogenesis and development of heart failure.

## 4. Lipolysis and FAO

Lipolysis is a catabolic process, involving the breakdown of TAGs into FAs and glycerol. Adipose triglyceride lipase (ATGL) and hormone-sensitive lipase (HSL) are key enzymes in lipolysis that generate FAs as energy substrates for FAO and subsequent ATP production. Fatty acid oxidation enzymes are located in the mitochondrial matrix, and FAs in the cytosol must be further activated and transported into the mitochondria. The first reaction of FA activation is catalyzed by ACS via the formation of fatty acyl-CoA, which is transiently attached to the hydroxyl group of carnitine by CPT1 to form fatty acyl-carnitine. Fatty acyl-carnitine ester diffuses into the inner mitochondrial membrane with the help of the acyl-carnitine/carnitine transporter. The final step of the carnitine shuttle pathway involves the regeneration of fatty acyl-CoA by carnitine acyltransferase II (CPT2), and carnitine reenters the intermembrane space via the acyl-carnitine/carnitine transporter. Fatty acyl-CoA then undergoes FAO to produce acetyl-CoA, which enters the TCA cycle and generates nicotinamide adenine dinucleotide (NADH) and FADH_2_ (flavin adenine dinucleotide) for the electron transport chain to produce ATP (Figure 1).

### 4.1. Intracellular LIPOLYSIS Enzymes ATGL and HSL

Patients with heart failure show high TAG levels in the left ventricular tissue [69]. Lipid accumulation can result from increased TAG synthesis or defective lipolysis. Adipose triglyceride lipase (ATGL) is a rate-limiting enzyme in intracellular lipolysis that releases FAs from TAG. Most patients carrying the ATGL mutation develop cardiac steatosis and present with cardiomyopathy [70]. In heart failure, most ATGL-deficient patients develop severe cardiac dysfunction that requires heart transplantation [71]. Cardiomyocyte-specific ATGL overexpression protects against cardiac dysfunction in a murine model of pressure overload-induced heart failure. The mice show reduced myocardial TAG content; however, the levels of DAG and ceramides remained unchanged. Notably, the FAO rate decreases, whereas the glucose oxidation rate increases [72]. In contrast, ATGL deficient mice develop excessive lipid accumulation, cardiac insufficiency, and lethal cardiomyopathy [73]. ATGL generates essential mediators that serve as ligands for PPAR activation. ATGL knockout mice treated with PPARα agonist completely restore normal heart function and prevent premature death [74,75]. Cardiac ATGL protein and TAG levels significantly increased in a murine model of diabetic cardiomyopathy, and ATGL deficiency results in lipotoxicity and diastolic dysfunction, whereas ATGL overexpression in cardiomyocytes is resistant to cardiac dysfunction [76]. These studies strongly indicate that ATGL activity affects cardiac function and that ATGL is a promising target for heart failure treatment.

Hormone-sensitive lipase (HSL) is the second lipolytic enzyme next to ATGL that catalyzes the breakdown of DAG into monoacylglycerol (MAG) and FAs. The substrates for HSL are much broader than that for ATGL, which including cholesteryl esters, retinyl esters, and TAG [77]. Humans with HSL deficiency show dyslipidemia, hepatic steatosis, systemic insulin resistance, and diabetes; however, no cardiac phenotype was found [78]. Murine studies have primarily focused on diabetic cardiomyopathy. Cardiac overexpression of HSL inhibits myocardial steatosis and fibrosis in diabetic mice by hydrolyzing toxic lipid metabolites [79]. Moreover, cardiac overexpression of HSL in ATGL-null mice rescues the cardiac dysfunction caused by ATGL knockout. This indicates that HSL compensates for ATGL deficiency [80]. HSL knockout mice have impaired adipose lipolysis and male infertility without severe cardiac defects. However, HSL-deficient mice are still protected from high-fat diet-induced cardiac insulin resistance associated with reduced intramuscular TAG [81]. The role of HSL in lipolysis may be more complex, and the elucidation of a more detailed mechanism requires further exploration.

Triacylglycerols are stored in the heart as lipid droplets. The presence of Perilipin proteins on the surface of lipid droplets protects TAG from lipolysis. It comprises five members: Perilipin 1–5. Perilipin 5 is abundantly expressed in the heart, Perilipins 1 and 4 are mainly expressed in adipose tissue, whereas Perilipin 2 and 3 are found in a variety of tissues. Compared to wild-type mice, Perilipin 5 knockout mice lacked detectable lipid droplets and more actively oxidized FA. Mutant mice show a greater decline in cardiac function with age [82]. However, type 1 diabetic Perilipin 5-null mice exhibit lower levels of lipotoxic molecules and resistance to diabetes-induced cardiac malfunction [83]. Mice that specifically overexpress Perilipin 5 in the heart have increased TAG content, and a chronic excess of lipid droplets causes mild heart dysfunction [84]. In addition, Perilipin phosphorylation by protein kinase A is required for the translocation of HSL from the cytosol to lipid droplets, a key event in triggering lipolysis [85]. However, cardiac overexpression of Perilipin 5 with a mutant phosphorylation site shows a comparable level of cardiac TAG compared to non-mutated Perilipin 5 transgenic mice, with completely normal heart function in these two mice. This finding suggests that Perilipin 5-mediated cardiac lipolysis requires multiple Perilipin 5 phosphorylation sites [86].

### 4.2. Extracellular Lipolysis Enzyme Lipoprotein Lipase (LPL)

The energy of the heart mainly relies on FA oxidation. Therefore, the supply of FAs to the heart is essential for cardiac function. The hydrolysis of TAG-rich lipoproteins by lipoprotein lipase (LPL) is another source of FAs in the heart in addition to the uptake of albumin-bound FAs derived from adipose tissue. Lipoprotein lipase-null mice exhibit severe hypertriglyceridemia, reduced high-density lipoprotein levels, and neonatal death [87]. LPL is expressed at its highest level in the heart, and its expression in the heart alone is sufficient to maintain normal plasma triglyceride levels and rescue LPL-null mice from neonatal death [88]. LPL overexpression in cardiomyocytes causes dilated cardiomyopathy with lipid accumulation in the heart [89]. In contrast, cardiac-specific LPL knockout mice have significantly elevated plasma triglyceride levels and increased glucose oxidation; however, the cardiac uptake of albumin-bound FAs is unchanged [90]. Cardiac LPL deletion mice died within 48 h of pressure overload, and older mice developed cardiac dysfunction [91]. Thus, LPL may play a central role in maintaining the balance of plasma TAGs during heart failure.

Recently, several LPL regulators were discovered, including lipase maturation factor 1 (LMF1), Sel suppressor of Lin-12-like 1 (SEL1L), glycosylphosphatidylinositol-anchored high-density lipoprotein-binding protein 1 (GPIHBP1), and angiopoietin-like proteins (ANGPTLs) [92]. LMF1 and SEL1L in endoplasmic reticulum functionally ensure the maturation of LPL [93]. The mature LPL will be captured by GPIHBP1 on the surface of endothelial cell, and then it moves to the endothelial lumen through transcytosis [94,95]. ANGPTL3, 4, and 8 have been reported to inhibit LPL activity and increase plasma triglyceride level in a tissue-specific manner [96,97,98]. Additionally, the ANGPTL3–ANGPTL8 complex has been shown to dramatically suppress the activity of LPL compared to either protein alone [99]. As ANGPTL3 is exclusively expressed in the liver, it is likely to be released as a complex with ANGPTL8 to suppress LPL activity in fat and muscle tissues. ANGPTL4 is expressed in numerous tissues and may act as a local LPL inhibitor [100]. Interestingly, transgenic overexpression of ANGPTL4 in the heart developed left-ventricular dysfunction. The mice exhibited hypertriglyceridemia after inhibiting LPL activity [101]. Importantly, patients in a dyslipidemia cohort showed association between LMF1 gene variants and postheparin LPL activity [102]. Mutation or deletion in the GPIHBP1 gene also resulted in severe hypertriglyceridemia in patients [103]. Moreover, people carrying loss-of-function variants in ANGPTL3 or ANGPTL4 have a reduced risk of coronary artery disease with decreased plasma level of triglycerides [104,105,106]. It suggests that the modulation of these LPL regulators could offer a therapeutic treatment for hypertriglyceridemia and reduce the risk for related heart disease.

### 4.3. Carnitine Acyltransferase I (CPT1)

Carnitine acyltransferase I (CPT1) is responsible for the formation of fatty acyl-carnitines by transferring the acyl group of fatty acyl-CoA to carnitine. It is also known as carnitine palmitoyltransferase I as its main product is palmitoylcarnitine. A decreased rate of FAO is suggested to occur secondary to the reduced carnitine content in the hypertrophied myocardium [107,108,109]. The role of propionyl L-carnitine in cardiac dysfunction was extensively investigated in the last century because it is a naturally occurring carnitine derivative that increases tissue carnitine levels [110]. Multiple studies show that the administration of propionyl-L-carnitine improves the contractile function of hypertrophied hearts [109,111]. However, the drug failed to normalize FAO, and the beneficial effect was due to the increased efficiency of ATP in cardiac function [112]. CPT1 is the rate-limiting enzyme in FAO, and it was actively investigated as a potential therapeutic target. CPT1 activity significantly decreases in hypertrophied hearts under pressure overload [113] and advanced heart failure [9]. CPT1 has three isoforms: CPT1a, CPT1b, and CPT1c. CTP1a is predominant isoform in the liver, and it is also known as a liver isoform. CPT1b is highly expressed in the heart and skeletal muscles, whereas CPT1c is only expressed in neurons. Heterozygous CPT1b knockout mice exhibit exacerbated cardiac hypertrophy and are susceptible to premature death due to congestive heart failure after pressure overload. Moreover, the mice presented with severe mitochondrial abnormalities and myocardial lipotoxicity, leading to cardiomyocyte apoptosis [114]. However, clinical trials show that specific CPT1 inhibitors (such as etomoxir) improve the cardiac function in patients with heart failure [115]. In contrast, overexpression of the CPT1 liver isoform elevates atrial natriuretic peptide levels [116]. This indicates that it induces hypertrophic signaling. CPT1 may be beneficial in the progression of heart failure; however, further studies are required to elucidate its role.

### 4.4. Fatty Acid Oxidation

Efficient energy production in cardiomyocytes requires three mitochondrial metabolic pathways: FAO, the TCA cycle, and the electron transfer chain. Acetyl-CoA produced from FAO enters the TCA cycle to generate NADH and FADH_2_ for the electron transport chain to produce ATP in normal hearts. Therefore, the rate of FAO is regulated by FAs supply and uptake, malonyl-CoA content, the ratio of FAD/FADH_2_ and NAD^+^/NADH, the mitochondrial acetyl-CoA/CoA ratio, and transcriptional and post-translational modifications of FAO enzymes [117]. Decreased FAO is reported in humans with idiopathic dilated cardiomyopathy [118,119,120,121], post-infarction heart failure [122], heart failure in Dahl salt-sensitive hypertensive rats [123], pressure overload-induced heart failure models [124,125,126], and canine models of cardiac pacing [127]. This alteration may be caused by a reduction in the number of genes and enzymes involved in FAO [9,124,126]. In contrast, FAO is highly induced in type 2 diabetes [128], obesity [129], and heart failure with a preserved ejection fraction [130]. Activated FAO also consistently occurs in diabetic and obese mice lacking leptin [131,132]. The increased FAs and induction of FAO enzymes by lysine acetylation may be the underlying mechanism [133,134]. 

Nicotinamide adenine dinucleotide (NAD) serves as the major electron carrier coenzyme in all three FAO stages. Initially, FAs undergo β-oxidation and remove successive two-carbon units to form acetyl-CoA. The formation of acetyl-CoA releases two pairs of electrons. NAD^+^ and FAD serve as electron acceptors and carry electrons in the form of NADH and FADH_2_. In the second stage, acetyl-CoA enters the TCA cycle to generate NADH and FADH_2_. Finally, NADH and FADH_2_ electron carriers produced in the above two stages donate electrons to the mitochondrial respiratory chain for ATP synthesis (Figure 1). Therefore, the NAD^+^/NADH redox pathway is critical for FA metabolism in patients with heart failure. A reduction in NAD or a decreased NAD^+^/NADH ratio is observed in patients with heart failure and in a murine model with pathologic hypertrophy [135,136]. Consistently, impaired expression of genes involved in NAD biosynthesis was confirmed in cardiac tissue from patients with ejection fraction preserved heart failure [137]. Mechanistically, a low level of NAD stimulates the hyperacetylation of mitochondrial proteins by inhibiting sirtuin, which impairs the cytosolic redox state and energy deficiency. Notably, the elevation of NAD^+^ levels by stimulating the nicotinamide phosphoribosyl transferase (Nampt)-dependent NAD^+^ salvage pathway or supplying NAD precursors improves cardiac function in response to stress (Figure 2) [135,136]. Furthermore, a clinical trial revealed that the administration of NAD precursor in patients with advanced heart failure enhances respiratory capacity through complex I activity in peripheral blood mononuclear cells and reduces proinflammatory cytokine gene expression [138]. 

The mitochondrial respiratory chain consists of five members (complexes I–IV) that are the engines producing energy in the heart. In contrast, complex I is considered a potential source of oxygen free radicals in the failing myocardium [139]. Therefore, determining the role of complex I in heart failure is challenging. Rong Tian et al. modeled the impairment of respiratory capacity by knocking out the Ndufs4 gene encoding a critical protein for complex I assembly. Cardiac deletion in Ndufs4 mice maintains normal cardiac function that is consistent with a >40% reduction in respiration. However, the heart develops accelerated dysfunction under pressure overload or repeated pregnancies. The authors thought that this phenotype was derived from a decreased NAD^+^/NADH ratio owing to complex I deficiency and increased protein acetylation [140]. Moreover, the knockout of Ndufs4 in macrophages worsens cardiac function 30 days after myocardial infarction. Suppression of respiration in macrophages impairs efferocytosis, leading to a low expression of anti-inflammatory cytokines and tissue repair factors [141]. The mitochondrial respiratory chain has multiple functions; these studies confirmed the cell type-dependent functions of these proteins in heart failure.

## 5. The Regulatory Factors of Fatty Acid Homeostasis

Transcriptional factors, such as PPARs and phosphorylation kinase (such as AMP-activated protein kinase (AMPK)), are central mechanisms that control complex metabolic networks in the heart during hemodynamic stress, as they can tune energy utilization of the heart by FA or glucose [4]. Transcription control requires three components: upstream events that activate signaling, molecular mechanisms that cooperate with transcription factors, and downstream actions that transcribe target genes [142]. Additionally, protein kinase also needs molecular triggering, as well as spatial and temporal factor cross-talking. Modification of these components has emerged as a potential target for therapeutic intervention.

### 5.1. Peroxisome Proliferator-Activated Receptors (PPARs)

Peroxisome proliferator-activated receptors are nuclear receptors sharing a common structure, including a conserved DNA-binding domain and a ligand binding domain [143]. They comprise three subtypes (PPARα, PPARδ, and PPARγ), and each is expressed in the heart. PPARα and PPARδ are relatively abundant [144]. PPARα expression is relatively decreased in heart failure patients with hypertensive heart disease compared to patients with normal cardiac function [145,146]. Consistently, PPARα is deactivated in ventricular pressure overload studies in mice, which is accompanied by the downregulation of FAO enzymes [147]. The activation of PPARα increases several downstream targets, including FAT/CD36, CPT1, and MCD [148]. PPARα deficiency in the heart is sufficient to maintain normal energy homeostasis and cardiac function. However, knockout mice show accelerated cardiac remodeling and contractile dysfunction during hemodynamic overload [149,150]. Transcriptome analysis reveals the activation of glucose metabolism genes and a switch in substrate utilization from FA to glucose. Notably, overexpression of glucose transporter GLUT1 in PPARα-deficient mice improves the metabolic and functional defects. This suggests that the upregulation of glucose metabolism from intrinsic responses to PPARα deficiency or from glucose transporter overexpression has different effects on heart failure. In contrast, PPARα activation by inducible transgenic mice maintains myocardial function with enhanced FAO in the early stages of heart failure [151]. PPARα knockout mice are protected from the severe cardiomyopathic phenotype in diabetes-induced cardiac hypertrophy. However, PPARα overexpression in diabetic hearts exacerbated the hypertrophy with myocardial triglyceride accumulation [152,153]. These results suggest that PPARα modulation could be a promising therapeutic strategy for heart failure.

PPARγ/δ is differentially regulated in the heart. Unlike the expression of PPARα, PPARγ is highly induced in heart failure. As PPARγ is a downstream target of hypoxia-inducible factor 1 (HIF1α), the activation of PPARγ induced by HIF1α stimulated the FA uptake and lipid accumulation. Ventricular HIF1a deficiency inhibits PPARγ expression and attenuates stress-induced pathological hypertrophy [154]. Besides, cardiac PPARγ overexpression develops a dilated cardiomyopathy associated with increased lipid stores [155]. The detailed mechanism is unclear; however, PPARγ-induced cardiolipotoxicity is ameliorated by deleting PPARα [156], and PPARγ activation seems to be protective in sepsis-related cardiac dysfunction [157]. PPARδ deletion downregulates the expression of FAO genes and decreases the basal myocardial FAO rate. These mice exhibit cardiac dysfunction and progressive lipid accumulation [158]. 

PPARγ coactivator 1 (PGC1) is defined as a protein that interacts with transcription factors and increases the probability of target gene transcription. PPARα, PPARδ, and PPARγ are all subject to transcriptional coactivation by PGC1. There are three PGC1 members: PGC1α, PGC1β, and PGC1-related coactivator [159]. PGC1α repression is a signature of the failing human heart [160]. PGC1 activity induces the expression of nuclear and mitochondrial genes in cultured cardiomyocytes, in addition to its role in cooperation with PPARs. However, cardiac-specific PGC1 overexpression develops dilated cardiomyopathy with uncontrolled mitochondrial proliferation [161]. Consistently, serious mitochondrial structural derangements were observed in the hearts of PGC1α/β-deficient mice during postnatal growth associated with the development of lethal cardiomyopathy. However, PGC1α/β-deficient adult mice did not result in heart failure and mitochondrial abnormalities [162]. PGC1α knockout mice develop marked dilated cardiomyopathy with the downregulation of several target genes in FA metabolism and electron transport chain in the pressure overload heart failure model [163].

### 5.2. AMP-Activated Protein Kinase (AMPK)

AMP-activated protein kinase serves as a cellular fuel gauge and metabolic regulator in the heart. It is a heterotrimeric complex, comprising a catalytic α subunit and two regulatory β and γ subunits. The α subunit contains a serine–threonine kinase domain that includes an activating Thr^172^ residue. Phosphorylation of this site is critical for AMPK activation [164]. AMPK regulates FA synthesis by phosphorylating ACC and HMG-CoA reductase [165]. It induces the expression and translocation of FAT/CD36 to the cell membrane and improves FA transport [166,167,168]. Furthermore, it regulates genes, such as PGC1, MCD, CPT1, and GLUT4 [169,170,171]. However, AMPK activation during pressure overload stress mainly increases the rates of glucose transport and glycolysis [172]. 

AMPKα2 knockout mice exacerbate ventricular hypertrophy and cardiac dysfunction in the pressure overload model [173] or high-fat diet-induced heart failure model [174]. In contrast, long-term activation of AMPK by the specific chemical activator AICAR attenuates pressure overload-induced cardiac hypertrophy [175]. In addition, AMPK deficiency in the ischemic heart increases cardiac injury and impairs contractile function with exacerbated ATP depletion [176,177]. Thus, AMPK exerts protective effects against heart failure. Several studies confirm that the pharmacological AMPK activator (metformin) improves ventricular function and survival in heart failure [178,179]. 

**Table 1 metabolites-13-00615-t001:** The classification and function of fatty acids.

Classification	Typical FAs	Carbon Skeleton	Common Name	Function
Saturated FAs	*n*-Dodecanoic acid	12:0	Lauric acid	Increased risk of cardiovascular disease by aggravating dyslipidemia [180]
	*n*-Tetradecanoic acid	14:0	Myristic acid	Increased risk of coronary heart disease [180]
	*n*-Hexadecanoic acid	16:0	Palmitic acid	Increased risk of coronary heart disease [180]
	*n*-Octadecanoic acid	18:0	Stearic acid	Increased risk of coronary heart disease [180]
	*n*-Eicosanoic acid	20:0	Arachidic acid	Decreased risk of incident heart failure [181]
	*n*-Teracosanoic acid	24:0	Lignoceric acid	Decreased risk of incident heart failure [181]
Monounsaturated FAs	*cis-9-* Hexadecanoic acid	16:1 (Δ^9^)	Palmitoleic acid	Increased risk of heart failure [59]
	*cis-9-* Octadecanoic acid	18:1 (Δ^9^)	Oleic acid	Not associated with heart failure risk [59]
Polyunsaturated FAs	*cis, cis*-9,12-Octadecadienoic acid	18:2 (Δ^9,12^)	Linoleic acid	Decreased risk of coronary heart disease [182]
	*cis, cis, cis*-9,12,15-Octadecatrienoic acid	18:3 (Δ^9,12,15^)	α-Linolenic acid	Increased risk of cardiovascular disease [183]
	*cis, cis, cis, cis*-5,8,11,14-Icosatetraenoic acid	20:4 (Δ^5,8,11,14^)	Arachidonic acid	Increased risk of cardiovascular disease [184]
	*cis, cis, cis, cis, cis*-5,8,11,14,17-Eicosapentaenoic acid	20:5(Δ^5,8,11,14,17^)	Eicosapentaenoic acid	Decreased risk of cardiovascular disease and incident heart failure [185,186]
	*cis, cis, cis, cis, cis, cis*-4,7,10,13,16,19-Docosahexaenoic acid	24:6 (Δ^4,7,10,13,16,19^)	Docosahexaenoic acid	Decreased risk of cardiovascular disease [185]

## 6. Conclusions and Perspectives

Emerging evidence supports the notion that unbalanced FA metabolism occurs in heart failure, which is characterized by the repression of FAO and the accumulation of toxic lipids. Almost all enzymes are downregulated in heart failure, including FAT/CD36, FATPs, ACS, MCD, and CPT1. However, FASN and SCD lipogenic enzymes are highly induced, thereby causing excess DAG or ceramide storage (Figure 3). Consequently, the heart suffers from a double burden of insufficient energy supply and lipotoxicity. ACC and ATGL may serve as potential targets to treat hemodynamic stress-induced heart failure. In contrast, FAT/CD36, DGAT, ATGL, HSL, and Perilipin may be candidates to treat cardiac dysfunction related to metabolic syndrome. Fatty acid metabolism involves a tight and cooperative network; therefore, transcriptional factors and protein kinase with the integrated capacity of several enzymes provide another potential therapeutic strategy in failing hearts. Importantly, clinical trials in patients with heart failure show that supplementation with NAD precursors has beneficial effects. The limitation of this review is we only focus on some key enzymes and important regulatory factors, and other proteins, such as cholesterol biosynthesis enzymes, are not involved here. Overall, significant progress has been made in broadening our understanding of the pathophysiology of cardiac dysfunction. However, further comprehensive studies are warranted to identify therapeutic targets to treat heart failure.

## Figures and Tables

**Figure 1 metabolites-13-00615-f001:**
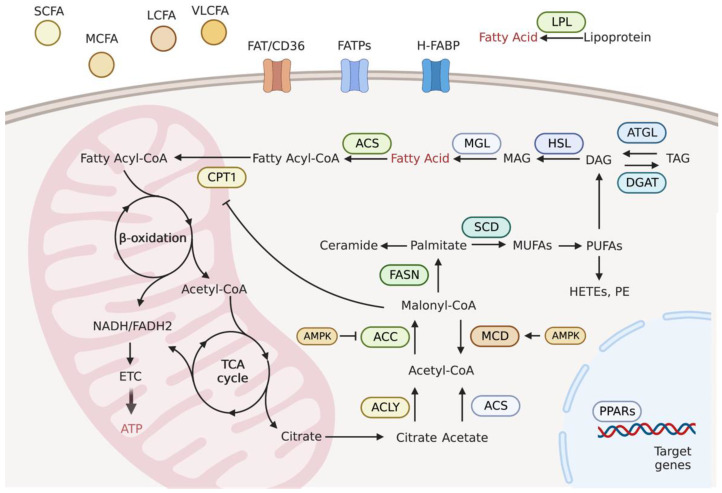
The cardiomyocytes obtain fatty acids (FAs) from exogenous uptake and de novo lipogenesis. The entry of FAs from microenvironment needs specific transporters, including FAT/CD36, FATPs, and H-FABP. The cellular FAs obtained from extracellular uptake or intracellular lipolysis can be used for ATP production through fatty acid oxidation in the mitochondria. De novo lipogenesis relies on citrate and acetate. With the help of ACLY, ACS, ACC and FASN, palmitate is ultimately generated, which is further desaturated and elongated to form other lipid species. The transcription factor PPARs with the integrated capacity of several enzymes in FA metabolism control the entire system. Abbreviations: SCFA, short-chain FA; MCFA, medium-chain FA; LCFA, long-chain FA; VLCFA, very long-chain FA; LPL, lipoprotein lipase; FAT/CD36, fatty acid translocase/cluster of differentiation 36; FATPs, fatty acid transport proteins; H-FABP, heart-type fatty acid-binding protein; ACS, acetyl-CoA synthetase; CPT1, carnitine acyltransferase I; ETC, electron transport chain; ATP, adenosine triphosphate; ACLY, ATP-citrate lyase; ACC, acetyl-CoA carboxylases; MCD, malonyl-CoA decarboxylase; FASN, fatty acid synthase; SCD, stearoyl-CoA desaturase; ATGL, adipose triglyceride lipase; DGAT, diacylglycerol acyltransferase; HSL, hormone-sensitive lipase; MGL, monoglycerol lipase; TAG, triacylglycerol; DAG, diacylglycerol; MAG, monoacylglycerol; PPARs, peroxisome proliferator-activated receptors; AMPK, AMP-activated protein kinase.

**Figure 2 metabolites-13-00615-f002:**
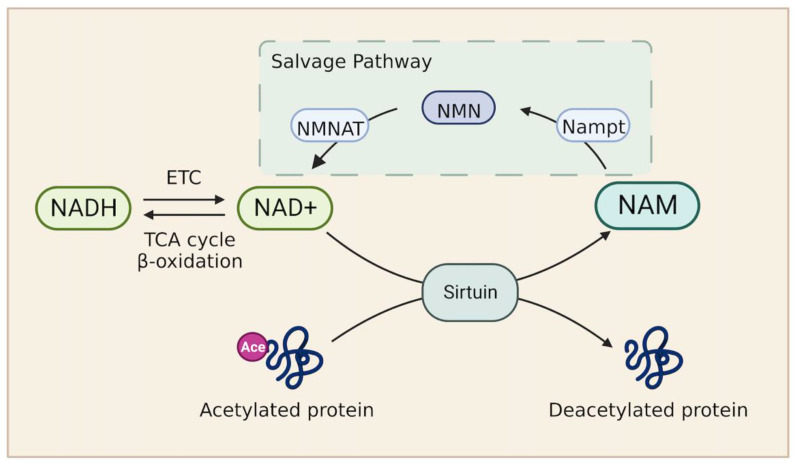
The nicotinamide adenine dinucleotide (NAD) metabolism. The level of NAD^+^ is determined by NAD^+^ synthesis from the salvage pathway or NAD^+^/NADH ratio. NAD^+^ is required in the Sirtuin-mediated deacetylase reaction. This reaction also generates NAM. NAM then enters the salvage pathway. Nampt is the rate limiting enzyme in this pathway, catalyzing the conversion from NAM to NMN. NMN is thereby converted to NAD^+^ by NMNAT. Abbreviations: NAD, nicotinamide adenine dinucleotide; Nampt, nicotinamide phosphoribosyl transferase; NAM, nicotinamide; NMN, nicotinamide mononucleotide; NMNAT, nicotinamide mononucleotide adenyltransferase.

**Figure 3 metabolites-13-00615-f003:**
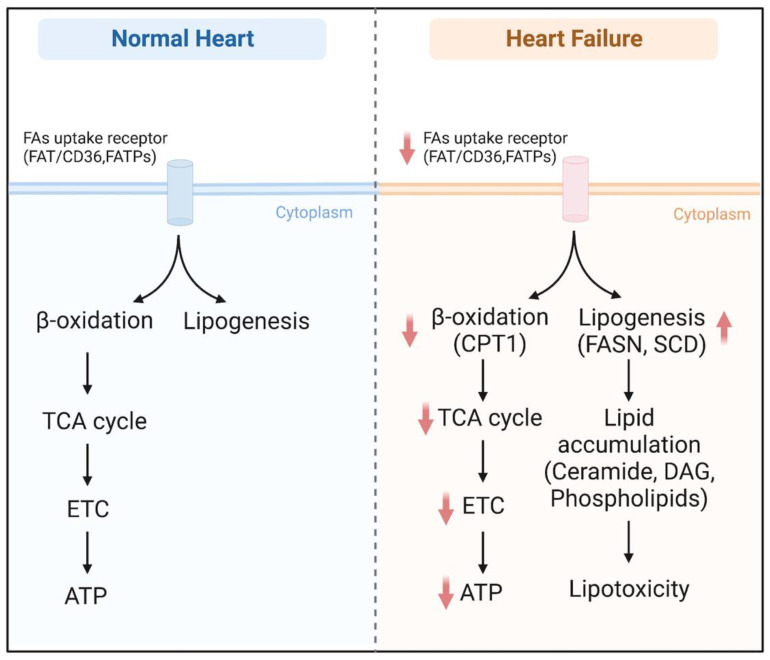
Overview of fatty acid metabolism in the normal heart and failing heart. Abbreviations: FAT/CD36, fatty acid translocase/cluster of differentiation 36; FATPs, fatty acid transport proteins; TCA cycle, tricarboxylic acid cycle; ETC, electron transport chain; ATP, adenosine triphosphate; FASN, fatty acid synthase; SCD, stearoyl-CoA desaturase; DAG, diacylglycerol.

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
