# Peer review of "Integrated Control of Fatty Acid Metabolism in Heart Failure"

_metabolites, 2023, doi:10.3390/metabo13050615_

Round 1

Reviewer 1 Report

This is a review article regarding fatty acid metabolism and heart failure, which reviewed fatty acid uptake, lipogenesis, lipolysis and FAO, and regulatory factors of fatty acid homeostasis. This reviewer considers that this review article was well written, but figures should be added for better understanding. This reviewer has some comments as described below. 

Major comments:

1.     Introduction section, lines 30-32. The authors indicated that fatty acids are divided into 4 groups. It should be also shown in Figure. 

2.     Figures 1-2 in the original version should be put in the text. 

3.     It is much better for readers to understand briefly. In this aspect, the authors should add more Figures for better understanding. 

Reviewer 2 Report

This is a well written and comprehensive review.

However, I have two  comments  referring to the paragraph 4.2  on lipoprotein lipase which in my opinion should be broadened with some information on  the interaction of LPL with angiopoietin -like proteins (ANGPTLs). It is well known that these proteins effect physiological variations of LPL activity and higher level of ANGPTL-4  was reported to be correlated with heart failure in human subjects. 

LPL is a rate-limiting enzyme for hydrolyzing TG, generating free fatty acids that are taken up by peripheral tissues including the heart, muscle and adipose tissue. Therefore the statements in the lines 315 and 317 – “cardiac specific LPL” should be modified as in my opinion they are misleading.

The main question addressed in this comprehensive  review refers to integrated regulation of fatty acid metabolism  with a particular focus on enzymes which may become  targets for future therapeutic strategies in patients with heart failure.

I think this topic is original and adds important information however,  as the authors mention at the end of Discussion the review has some limitations. In my opinion the authors could broaden the section describing the regulatory proteins of  lipoprotein lipase function.

Figures are informative, in particular the overview on  Fig.2 is very helpful to understand changes in metabolism of fatty acids in heart failure.

Reviewer 3 Report

The review manuscript authored by Li, et al unravels that fatty acid metabolism is associated with chronic heart failure. It may have translational implications and provides targets tackling heart failure. 

Major concerns

1. There are too many abbreviations in this manuscript. Although many abbreviations were provided with full names in Fig. 1 it will be convenient to readers to have all abbreviations at the beginning of this manuscript. 

2. Authors must indicate the place for Figures 1 and 2 in the text body and explain these two figures. Both Fig. 1 and Fig. 2 were not listed in the text. 

3. AMPK is a critical enzyme in fatty acid metabolism. It should be added into Fig. 1.

4. As described in Figure 1, endogenous lipogenesis is mainly synthesized within cells. Do fatty acid membrane transport proteins affect lipogenesis? It needs to be clarified in Figure 2 and text. 

Minor concerns:

None

Round 2

Reviewer 1 Report

This reviewer has no further comment.